# The Study of Chemical and Thermal Influences of the Environment on the Degradation of Mechanical Properties of Carbon Composite with Epoxy Resin

**DOI:** 10.3390/polym14163245

**Published:** 2022-08-09

**Authors:** Tatiana Kojnoková, František Nový, Lenka Markovičová

**Affiliations:** 1Department of Materials Engineering, University of Žilina, 010 26 Žilina, Slovakia; 2Research Centre, University of Žilina, 010 26 Žilina, Slovakia

**Keywords:** carbon composite with epoxy resin, chemical and thermal influence, glass transition temperature, mechanical properties, material surface macrostructure

## Abstract

The present research deals with the investigation of the influence of aqueous solutions of chemical substances in combination with temperature on the change of material properties of polymer composites based on epoxy resins reinforced with carbon fibers. The aim of the research was to investigate and evaluate the impact of degradation processes due to the influence of chemical environments of different temperatures and time of their action on changes in the material properties and macrostructure of carbon composite with epoxy resin. The chemical and thermal influence of the environment on the experimental material was evaluated by monitoring changes in mechanical properties, glass transition temperature, and material surface macrostructure. The achieved results show different behavior of the experimental composite material in different environments, while it was demonstrated that the degradation effect of chemical and thermal influences on changes in material properties increase with increasing temperature. Among the investigated environments (NaCl, NaOH, and H_2_SO_4_), exposure to 10% NaOH, and 15% H_2_SO_4_ had the greatest degradation influence on the polymer composite, and exposure to 20% NaCl had the smallest influence, which is also confirmed by invisible changes in material surface macrostructure and decrease of tensile strength by about 20%. Exposure to 10% NaOH resulted in significant surface roughening of the epoxy resin, white deposit creation on the surface, and a decrease of tensile strength by 35%. Opposite that, exposure to 15% H_2_SO_4_ resulted in the highlighting of the fiber yarns of the carbon fiber fabric, yellowing of the surface, surface pore occurrence, and a decrease of tensile strength by 35%.

## 1. Introduction

This submitted article shows changes in material surface macrostructure and thermo-mechanical properties of the carbon fiber-reinforced polymer (CFRP) composite plates after exposure to the influence of chemical solutions which present aggressive conditions to which the material can be exposed in real conditions. Many basic studies in aggressive conditions have been done on metallic materials but not much has been done on the properties of the CFRP composite for specific environmental conditions [1,2,3,4,5].

Polymer-based composite materials are currently among relatively frequently used structural materials. Composites are materials made of two or more components with a different material interface. The materials used to make the composite have significantly different physical and chemical properties. The combination of properties creates a material that differs in properties from the original materials used. Composites are considered a substitute for traditional materials used mainly in the automotive and aerospace industries. An important characteristic of the composite is that the technical properties required of the material can be achieved in the final product by a consistent selection of matrix and reinforcement [6,7]. Composites with a polymer matrix are used due to their excellent properties. Fiber-reinforced polymers (FRP) dominate with their stiffness and increased strength compared to unreinforced polymers. By incorporating fillers, metal particles, or fibers, better mechanical properties are achieved. Compared to conventional materials, polymer composites are highly susceptible to degradation due to heat and humidity in demanding operating environments [8]. For example, the degradation changes after placing a composite in boiled water for a few hours are equal to degradation changes after the water immersion test at 50 °C for 200 days [9]. It is necessary to take into account the influence of environmental factors, such as temperature, humidity, and corrosive environments, as they affect the material properties of composites. The influence of increased temperature is visible in the decrease of modulus of elasticity and strength due to thermal softening. The work is focused on the study of changes in material properties of carbon composite with epoxy resin. These changes can occur after the chemical and thermal influence of the environment on the material, and thus the degradation of the material itself can occur, as was also studied in many previous types of research [9,10,11,12,13,14,15,16,17,18]. When the composite is exposed to moisture or a liquid environment, many polymer matrix composites absorb moisture by diffusion through the matrix (surface absorption). Analysis of moisture absorption shows that in the case of epoxy and polyester matrix composites, the moisture concentration initially increases with time and approaches the equilibrium level (saturation) after several days of exposure of the material to a humid environment. Examination of composites due to increased temperature and moisture absorption is called hygrothermal analysis [19].

Literature review:

Changes in polymer composites in various environments are influenced by the mutual chemical reactivity of the polymer matrix with the surrounding environment and the rate of diffusion. Diffusion can occur as a physical process in which macromolecules are moved apart by the diffusing substance and the polymer swells under the influence of the fluid softening and become less rigid, which can lead to cracks at the matrix/reinforcement interface. The formation of cracks can cause a violation of the connection of the fibers, thus allowing the entry of various aggressive liquids causing the degradation of the material [10]. Despite their high stability, composites with a polymer matrix can undergo degradation during long-term exposures to aggressive environments. The influence of chemicals on the degradation of composites can be manifested by a decrease in mechanical properties and a change in weight or appearance. In the research of [11], it was found that immersion of the CFRP composites in an alkaline environment resulted in a greater decrease in mechanical properties than immersion in an acidic environment. The flexural strength decreased after 36 days by 22% after exposure to 10% NaOH and by 16% after exposure to 10% HCl. The flexural modulus decreased by 27% after the influence of 10% NaOH and by 22% after the exposure to 10% HCl. Roughness increased with exposure time and was higher for samples immersed in an alkaline solution. In another work, Amaro [12] studied the influences of 10% HCl and 10% H_2_SO_4_ on the flexural properties of the CFRP composites and found that the flexural strength decreased by 16% after the exposure to HCl and 11% after the exposure to H_2_SO_4_ aqueous solution.

In the research of Mahmoud [13], the influence of exposure time on the flexural strength, hardness, and impact resistance of glass fiber-reinforced polymer (GFRP) composites was studied. It was found that the flexural strength was not changed within 30 days of immersion in 20% HCl aqueous solution, and after this period there was observed a 10% decrease in flexural strength. In terms of hardness, it was shown that the hardness decreased by approximately 15% after 90 days of exposure. During the first 60 days of exposure, the Charpy impact strength decreased by 5%, but between 60 and 90 days of exposure, a significant decrease of 10% was noted. A short literature review of research that was carried out on the influence of chemical environments and humidity on composites is shown in Table 1.

In the research of Uthaman [15], the influence of exposure to 5% HCl and 10% NaOH on the CFRP composites was investigated, and it was found that after 40 and 80 days at a temperature of 60 °C, the degradation of the material increased, mainly due to exposure to 5% HCl. It was concluded that during hygrothermal aging, the immersion solution significantly affects the mechanical properties, as the CFRP composites behaved differently in water, 10% NaOH, and 5% HCl. The changes were attributed to a cross-linking effect at higher temperatures. Tensile strength decreased by 20% with water, 25% with HCl, and 24% with NaOH aqueous solution. From the obtained results, it can be seen that the decrease in mechanical properties is higher in an acidic environment. 

It cannot be clearly stated that due to the influence of an alkaline environment, there is a higher decrease in mechanical properties compared to an acidic environment, as was observed in the work of Mortas [26]. The effect of the environment deeply depends on the used concentration of acid or alkaline. However, with the comparison of the same concentration of acid and alkaline [11], a higher decrease in material properties after exposure to an alkaline environment was observed.

Compared to polyesters, epoxy resin has better mechanical properties and higher moisture resistance. Nevertheless, the epoxy resin shows a decrease in mechanical properties due to the influence of applied humidity. Low humidity enables cross-linking, which improves mechanical properties. During degradation, polymers absorb moisture from the environment due to the high number of polar hydroxyl groups. Absorbed moisture causes significant volume expansion during reaction and loss of tensile, compressive, and shear strength. Moisture absorption can be minimized by proper storage and handling of the composite semi-finished product-prepreg. Semi-finished materials in the form of prepregs are stored in freezers and are therefore often exposed to condensation. The purity of the resin plays an important role in minimizing moisture. Consideration should be given to the purity and nature of the resin impurities; as cleaner resins are less susceptible to moisture. It follows that resins with higher purity have a longer shelf life [27]. Absorption of water by the resin can change the properties of the resin and the glass transition temperature of polymers. The stiffness of the composite may deteriorate if the glass transition temperature is lower than the moisture absorption temperature. Absorbed water causes the resin to swell. There is swelling of the resin matrix in the composites around the fiber. Even at room temperature with 95% relative humidity, epoxy plasticizes and swells. This reduces the residual compressive strength at the fiber/matrix interface caused by shrinkage during curing. Mechanical stresses are released between the fiber and the matrix. Voids present at the fiber/matrix interface or in the composite layer can cause plasticization of the resin. Water trapped in the cavities causes the appearance of blisters. The penetration of moisture into the epoxy resin leads to mechanical, chemical, and thermophysical changes [28].

In contrast with the composite literature, there are far fewer investigations that focus on the thermal and chemical influences on the degradation of mechanical properties of carbon fiber-reinforced polymer matrix composites [29,30]. To provide more information about the short-term behavior of CFRP composites in aggressive industrial environmental conditions, this study investigates the synergistic effect of environment and temperature on the degradation of the CFRP composite produced from prepreg. The water solutions with higher concentrations (20% NaCl, 10% NaOH, and 15% H_2_SO_4_) and elevated temperatures (23, 40, and 60 °C) were used to shorten the exposure time. The degradation changes of mechanical properties were studied in the time period of three, six, and nine weeks using a tensile test, three-point bending test, and dynamic mechanical analysis. Light microscopy and scanning electron microscopy (SEM) were used to investigate the degradation surface changes.

## 2. Experimental Material and Methods

### 2.1. Experimental Material

For the study of chemical and thermal influences on the degradation of the CFRP composites, a hardened prepreg consisting of carbon fabric pre-impregnated with epoxy resin and labeled CFRP composite was used as experimental material. The samples were manufactured from commercially available carbon epoxy pre-impregnated rolls which were cut in required orientations and laid up by hands on the mold. The preparation flow chart and product diagram of the CFRP composite is shown in Figure 1. The autoclave production process followed [7].

Detailed characteristics of the experimental CFRP composite are shown in Table 2.

The experimental material is mainly used in the automotive industry, but in addition to automotive parts, various covers, protectors, or even panels are made from it. The experimental material used for research purposes in this study comes from the specialized Slovak company R&D Mold Machining, engaged in the production of composite pre-engaged laminated products. Production procedures used in the preparation of samples of experimental material are subject to the rules of protection of the “know-how” of the company. Due to the complicated shape of the slightly curved components produced as part of the company’s standard production, pre-impregnated laminate boards prepared simultaneously with real components were used for our experiments. By using the same prepregs, storage systems, and technological parameters in the autoclave, it was possible to prepare a homogeneous experimental material with a constant thickness with the same parameters as real products. The prepregs were cut into pieces in the desired cut orientations and subsequently laminated to a metal plate. They were then cured in an autoclave while observing the standard technological parameters of the production process. After the hardening process, the Department of Materials Engineering of the University of Žilina prepared samples with specific dimensions for the given tests. The experimental material consisted of three layers of prepregs with an orientation of 0/90°, ±45°, and 0/90°, which were pre-impregnated with epoxy resin with a total proportion of resin of 42 wt. %. The GG280T carbon fiber fabric, which is pre-impregnated with DT121HT epoxy resin, was used in all layers. The technical data of the GG280T carbon fiber fabric are twill weave 4 × 4, fiber area weight (FAW) 280 g/m^2^, fiber yarn type HS–3K (marking HS–high strength type of carbon fiber. Carbon fibers or filaments are bound in a fiber yarn which is identified by the number of carbon fibers they contain. The designation K stands for thousand, so a 3K fiber yarn is made of 3000 carbon fibers), warp count 7 th/cm, weft count 7.0 th/cm, and the thickness of the prepreg is 0.32 mm [7].

### 2.2. Exposure Conditions

In technical practice, degradation changes of carbon composites with epoxy resin occur mostly in the temperature range of working temperatures of 20–90 °C. Some standards can be used in specific industrial conditions; for example, for testing of the CFRP composite in seawater and sea sand concrete (SWSSC) conditions, but no international standards describe the testing conditions of the CFRP composite in alkaline and acidic water solutions. Many different testing conditions of immersion tests (duration, concentration, and temperature) can be found in the literature. On the basis of the demands of the CFRP composite producer and the short literature review [9,10,11,12,13,14,15,16,17,18], the exposure conditions listed in Table 3 were used.

Firstly, observation of the macrostructure surface was performed and subsequently, the CFRP composite was mechanically tested.

### 2.3. Methods

#### 2.3.1. Macroscopic Observation of the Surface Changes

The material surface macrostructure before and after exposure to different chemical environments at different temperatures was monitored using light microscopy using a Leica S9D stereo microscope (Leica Microsystems, Wetzlar, Germany) on samples with dimensions of 20×20 mm.

Scanning electron microscopy (SEM) was used for the study of surface changes at higher magnifications. When a bundle of primary electrons interacts with the surface of the body, secondary electrons are released, and they are subsequently detected. By default, SEM analysis is performed in a vacuum. The detected secondary electrons provide information on the surface morphology, chemical composition, and crystal structure of the material under investigation. A Nova NanoSEM 450 electron microscope (FEI, Hillsboro, OR, United States) was used to observe the surface of the CFRP composites.

#### 2.3.2. Tensile Test

The tensile test was based on the ASTM D3039 standard by using rectangular samples. The tensile strength of the samples of the CFRP composite was evaluated from the average of 5 samples for each series. Following sample dimensions for each series of samples were used such as a length of 150 mm, a width of 20 mm, and a thickness of 1 mm. The ends of the samples were covered with glued aluminum plates to avoid possible breakage of the samples by the jaws of the device thus causing unwanted tearing of the sample at the point of attachment to the jaws of the INSTRON 5985 universal testing device (INSTRON, Norwood, MA, United States). The samples were tested under ambient air temperature and constant testing speed of 0.00004 mm/min.

#### 2.3.3. Three-Point Bending Test

Bending properties were determined using the M350-5CT universal testing device (Testometric Co. Ltd., Rochdale, United Kingdom). The measurement was performed as a three-point bending test according to the EN ISO 178 standard for polymer materials. The distance of the supports was 32 mm, and the test loading speed was 2 mm/min. The diameter of the load mandrel was 14 mm, and the diameter of the lower supports was 10 mm. The measurement was carried out using the Testometric winTestAnaLysis 5.030 software. Rectangular samples with a length of 75 mm, a width of 20 mm, and a thickness of 1 mm were used. The value of the flexural strength and flexural modulus were taken from the average of 5 samples for each series. A force measuring head with a maximum force capacity of 1000 N was used. The sample was loaded until the break.

#### 2.3.4. Dynamic Mechanical Analysis

In the study of the experimental material, the dynamic mechanical analysis (DMA) was used as a three-point bending arrangement and the measurement was performed on a Mettler Toledo instrument (Mettler Toledo, Langacher, Switzerland) using the DMA STARe System software. The distance between the supports was 40 mm. The test samples were stressed at a constant frequency of 5 Hz with an amplitude of 100 μm. The test took place at a temperature interval of 25–200 °C with a heating rate of 3 °C/min. Rectangular samples with a length of 50 mm, a width of 10 mm, and a thickness of 1 mm were used. Samples were tested after the longest exposure time, namely 9 weeks. The value of the complex modulus of elasticity was taken from the average of 3 samples for investigated series. The modulus Eʹ, E’’ and the tan δ were measured as a function of temperature.

## 3. Results

### 3.1. Macroscopic Observation of the Surface Changes

By observing the surface of the experimental material using light microscopy at a magnification of 20×, the greatest visual changes can be seen after nine weeks of exposure. Changes in the macrostructure of the material surface after nine weeks of exposure at a temperature of 60 °C are documented in Figure 2.

The quality of the surface of the reference (unexposed) material is very good thanks to the correctly chosen technological parameters of the production process, after which the final CFRP composite contains only a small amount of surface imperfections (shallow scratches caused by subsequent careless handling). These surface imperfections do not affect the mechanical properties of the CFRP composite. 

Depending on the increase in the exposure temperature of the chemical environment, relatively significant surface changes can be observed on the surface of the investigated materials after exposure. The biggest differences are observable after exposures at a temperature of 60 °C.

The character of the surface of the material after exposure to 20% NaCl is very similar to the character of the surface of the unexposed material. Exposure to 10% NaOH resulted in significant surface roughening of the epoxy resin and white deposits of hydroxide and a small number of pores occurring on the surface. Exposure to 15% H_2_SO_4_ of the CFRP composite resulted in the highlighting of the fiber yarns of the carbon fiber fabric and a change in surface color. The accentuation and emergence of the fiber yarn of the carbon fiber fabric were caused by the degradation of the surface layer of the epoxy resin, which occurred through a partial chemical reaction that promoted the degradation of the material. The transparent, colorless resin turned yellow, especially at the interface of the fiber yarns.

The degradation changes of the surface of the CFRP composites after chemical and thermal exposure were also investigated using scanning electron microscopy. Degradative changes were investigated at the location of the fiber yarns. As a result of increasing the exposure temperature of the chemical environment, increasing surface differences can be observed. The biggest differences are observable at a temperature of 60 °C. It was confirmed that the biggest changes after exposure to individual chemical environments are most visible after nine weeks of exposure. Changes in the macrostructure of the material surface are documented in Figure 3. 

From the macroscopic observation, it was found that the greatest chemical and thermal influence on the materials was seen after nine weeks of exposure to 10% NaOH and 15% H_2_SO_4_ at a temperature of 60 °C. The character of the surface of the material after exposure to 20% NaCl is very similar to the character of the surface of the unexposed material. Exposure to 10% NaOH resulted in greater surface roughening of the epoxy resin. After exposure to 10% NaOH, white deposits of hydroxide and a small number of pores can be seen on the surface. After exposure to 15% H_2_SO_4_, an increased number of pores on the surface and a narrowing of the spaces between the fiber yarn that were previously filled by the resin can be seen, which was manifested by the fusion of the fiber yarn.

### 3.2. Tensile Test

To evaluate the influence of environmental conditions on the degradation of the CFRP composites, the conditioned samples were tested in tension. The tensile test was performed according to ASTM D3039. The mechanical properties of the CFRP composite were evaluated based on a tensile test before and after chemical and thermal exposures. Average tensile strength values are processed from the measurements. 

After the chemical and thermal exposures of the CFRP composite, the tensile strength values were evaluated. A decrease in tensile strength was recorded due to the influence of all selected environments. The CFRP composite showed the highest tensile strength value after three weeks of exposure to 20% NaCl at 23 °C (509.11 MPa). This value represents an 8.33% decrease in the tensile strength compared to the reference (unexposed) material (555.38 MPa). The lowest tensile strength value was observed after exposure to 20% NaCl at 40 °C (440.86 MPa), which represents a 20.62% decrease. The CFRP composite after six weeks of exposure showed the highest value of tensile strength after exposure to 15% H_2_SO_4_ at 60 °C (539.27 MPa). This value represents a 2.90% decrease in tensile strength. The lowest tensile strength value was observed after exposure to 10% NaOH at 60 °C (360.26 MPa), which represents a 35.13% decrease. The CFRP composite after nine weeks of exposure showed the highest tensile strength value after exposure to 20% NaCl at 40 °C (456.35 MPa). This value represents a 17.83% decrease in tensile strength. The lowest tensile strength value was observed after exposure to 15% H_2_SO_4_ at 40 °C (361.95 MPa), which represents a 34.83% decrease.

As a result of the CFRP composite exposure to 20% NaCl aqueous solution (Figure 4), the lowest tensile strength values were recorded at a temperature of 60 °C.

Due to the influence of the CFRP composite exposure to 10% NaOH (Figure 5), the greatest decrease in tensile strength after six weeks at 60 °C and the smallest decrease in tensile strength after six weeks at 40 °C were recorded.

Due to the influence of the CFRP composite exposure to 15% H_2_SO_4_ (Figure 6), the largest decrease in tensile strength was recorded after nine weeks of exposure and the smallest decrease in tensile strength after six weeks at 60 °C.

The typical experimental stress-strain curves of the CFRP composite immersed in 20% NaCl, 10% NaOH and, 15% H_2_SO_4_ at 60 °C for three, six and nine weeks can be seen in Figure 7.

### 3.3. Three-Point Bending Test

The three-point bending test was performed according to the EN ISO 178 standard. The mechanical properties, flexural strength, and flexural modulus, of the experimental material, were evaluated based on the three-point bending test before and after exposure to degradation environments. The CFRP composite after three weeks of exposure showed the highest flexural strength value after exposure to 15% H_2_SO_4_ at 23 °C (783.34 MPa) and 20% NaCl at 40 °C (782.99 MPa). These values were higher than the flexural strength values of the reference (unexposed) material (775.16 MPa), which represents a 1% increase in flexural strength. The lowest flexural strength value was observed after exposure to 10% NaOH at 60 °C (564.87 MPa), which represents a 27.13% decrease.

The CFRP composite after six weeks of exposure showed the highest flexural strength value after exposure to 20% NaCl at 60 °C (813.67 MPa). This value was higher than the flexural strength value of the unexposed material by 4.97%. The lowest value of flexural strength was observed after exposure to 10% NaOH at 60 °C (542.89 MPa) and 15% H_2_SO_4_ at 60 °C (704.33 MPa), which represents a subsequent decrease of 29.97% and 9.14%.

The CFRP composite after nine weeks of exposure showed the highest flexural strength value after exposure to 20% NaCl at 23 °C (808.77 MPa) and 10% NaOH at 23 °C (790.23 MPa). These values were higher than the flexural strength values of the unexposed material, which represents a 4.34% and 1.94% increase, respectively. The lowest flexural strength value was observed after exposure to 10% NaOH at 60 °C (542.98 MPa) and 15% H_2_SO_4_ at 60 °C (690.81 MPa), which consequently represents a 29.96% and 10.88% decrease. 

As a result of exposure to 20% NaCl aqueous solution (Figure 8), no visible changes in flexural strength were recorded (highest value after six weeks at 60 °C). The flexural modulus increased after three and six weeks with increasing temperature. After nine weeks, the flexural modulus values are almost the same at all temperatures.

As a result of exposure to 10% NaOH (Figure 9), a decrease in flexural strength with increasing temperature (except temperature 40 °C) was recorded. The lowest flexural strength values are after exposure to 10% NaOH at a temperature of 60 °C and the highest value is after nine weeks of exposure at 23 °C. After exposure to 10% NaOH, the highest values of the flexural modulus were measured after nine weeks of exposure.

As a result of exposure to 15% H_2_SO_4_ (Figure 10), the flexural strength decreased with increasing exposure time at a temperature of 60 °C (highest value after three weeks at 23 °C). The values of the flexural modulus were higher after three and nine weeks of exposure compared to the unexposed material. After six weeks of exposure, a slight decrease in the flexural modulus was noted.

### 3.4. Dynamic Mechanical Analysis

DMA analysis was performed on the samples of reference (unexposed) and exposed material. The given results represent the rate of degradation of the CFRP composite due to the action of selected chemical environments and temperatures after nine weeks of exposure. From the results of these measurements, the viscoelastic properties of the experimental CFRP composite were obtained, providing information on how individual chemical and temperature exposure conditions affected their material properties. Continuous measurement enabled the characterization of parameters complex modulus of elasticity E* and tan δ of the CFRP composite.

Figure 11, Figure 12 and Figure 13 show the DMA characteristics of the CFRP composite evaluated depending on the influence of individual chemical environments at the same temperature. At the same temperature of a different chemical environment, almost the same course of DMA dependencies is observed. The biggest change is observed for the material exposed to 10% NaOH and 15% H_2_SO_4_ at a temperature of 60 °C.

From the graphs processed from the measurements, it is clear that the influence of chemical environments and increased temperature leads to a decrease in the complex modulus of elasticity and a shift in the tan δ to lower values, as well as a significant decrease in the glass transition temperature for both investigated materials. The glass transition temperature T_g_ of the CFRP composites was determined from the dependence of the complex modulus of elasticity E* on the temperature T concerning the mechanical properties of the material (Table 4). T_g_ values were obtained from the intersections of two tangents of the curve leading from the vitreous region to the region of the viscoelastic state of the material.

In the case of the CFRP composite, a decreasing trend of T_g_ values is observed depending on the increasing temperature of the given chemical environment. Overall, the highest T_g_ value is achieved due to exposure to 20% NaCl at 23 °C. The lowest value of the glass transition temperature of the CFRP composite was measured after exposure to 10% NaOH at a temperature of 60 °C, which represents a 23.75% decrease in the T_g_ value compared to the reference (unexposed) CFRP composite. The glass transition temperature T_g_ of the experimental material was also determined from the dependence of the tan δ on the temperature T concerning the internal energy of the material (Table 5). T_g_ values in this case were obtained from the maximum peak of the curve. The CFRP composite shows two peaks only in samples affected by all three chemical environments at a temperature of 60 °C.

Compared to the T_g_ values obtained from the curves of the dependence of the complex modulus of elasticity on temperature, higher T_g_ values were determined by the curves of the dependence of the tan delta on temperature.

## 4. Discussion

Based on the observation of the material surface macrostructure using light microscopy, deterioration of the quality of the surface of the epoxy resin and changes in the interface between the fiber yarns were observed for the CFRP composite. In addition, a change in the color of the material was visible, similar to the work of Uthaman [15]. The chemical environment of 10% NaOH began to react with the epoxy resin, causing a chemical reaction on the surface of the material and the subsequent formation of reaction products.

Scanning electron microscopy showed surface degradation of the exposed CFRP composites. Similar nature of the surface degradation of the CFRP composites was also noted in the work of Ji [31], who noted the degradation of epoxy resin mainly due to the influence of HCl, while the acid caused more damage at a temperature of 60 °C than at a temperature of 23 °C. Similarly, exposure to 15% H_2_SO_4_ resulted in intense fiber approximation and fusion of the fiber yarn itself, eliminating the interface between fibers. Damage to the composite caused by plasticization and hydrolysis caused densification of the space between the fibers at a higher temperature. This was due to the contact of the solution with the surface of the fibers, which not only attacked the epoxy resin but also slightly attacked the carbon fibers themselves.

The chemical and thermal influences of the environment on the degradation of carbon composites with epoxy resin were studied and characterized based on changes in material properties. According to Dotson [32], in technical practice, degradation changes of carbon composites with epoxy resin occur mostly in the temperature range of working temperatures of 20-90 °C. Therefore, the experimental temperatures were selected from this temperature range. Based on current knowledge and results obtained from experimental measurements, it is clear that different chemical environments and temperatures have a significant influence on the CFRP composite itself, specifically on its structure, weight, mechanical properties, and surface quality. In the studied CFRP composite, there were changes in the polymer chain cleavage or cross-linking. The quality of the surface also changed due to the selected exposure conditions. Despite the observed changes in material properties due to various exposure conditions in selected environments, the CFRP composites retain their stability in less aggressive environments. The achieved results were compared with the research results of other works. According to Al-Ghuilani and Hammami [16], the degradation of the CFRP composites occurs in two stages. In the first phase, the epoxy resin is attacked by hydrogen diffusion. In the second phase, the fiber itself is attacked and cracks appear on the surface of the fiber, therefore the weakened adhesion (connection) of the fiber and the matrix leads to a weakening of the material and thus to a reduction in the mechanical properties of the composite. When an epoxy resin composite material is exposed to a hygrothermal environment (a combination of moisture and temperature) the T_g_ usually decreases and therefore the operating temperature of the material changes. This T_g_ modification reflects the degree of plasticization of the resin and the water/resin interaction in the material. The mechanical properties of the CFRP composites exposed to the chemical and thermal influences of various environments as part of the solution of this study in most exposure conditions decreased compared to the mechanical properties of the basic unexposed material, similar to the work of Uthaman [20] and Ji [31]. Epoxy resins generally show good mechanical and thermal properties, high resistance to chemicals, and heat and corrosion resistance Muñoz [17]. Among the selected chemical environments, 10% NaOH and 15% H_2_SO_4_ had the greatest influence on the mechanical properties of the CFRP composites. It cannot be clearly stated that due to the influence of an alkaline environment, there is a greater decrease in mechanical properties compared to an acidic environment, as was observed in the work of Mortas [26]. The decrease in the tensile strength and flexural strength of the CFRP composite is reflected in Table 6.

Changes in the complex modulus of elasticity E*, the tan δ, and the glass transition temperature T_g_ were identified by dynamic mechanical analysis in bending in a wide range of temperatures. As a result of nine weeks of exposure to selected chemical environments and temperatures, the T_g_ value decreased. Such behavior of the CFRP composites has been observed by several authors. For example, in the work of Zhou [18], a decrease in T_g_ was recorded due to the exposure of the CFRP composites to hygrothermal influence (a combination of moisture and heat). In our experiments, as a result of exposure to selected chemical environments at different temperatures, there was a decrease in the value of the complex modulus of elasticity, a shift in the tan δ to lower values, and a decrease in the glass transition temperature. The reason is the cleavage of the polymer chain, which was caused by exposure to 10% NaOH and 15% H_2_SO_4_, which represent relatively aggressive environments. The acid and hydroxide disrupted the interface between the carbon fiber fabric and the epoxy resin, which disrupted their adhesion due to the increased mobility of the polymer chains Ji [31]. The reduction of tan delta curves is greater at elevated temperatures than at lower temperatures of chemical environments. This might be due to the participation of the protons of the alkaline and acidic water solutions in chemical reactions between water molecules via hydrogen bonding and within the hydrophilic groups of the epoxy, which resulted in the generation of a less-plasticized region dispersed across the surface of the polymer [15]. According to previous studies, the irreversible degradation of aged materials with epoxy resin is due to moisture or water uptake, while plasticization occurs [33], and results in a decrease in the glass transition temperature [23,34]. Delasi [35] proposes that the state of absorbed water has a significant effect on the associated variation in T_g_. On the contrary, water contained in clusters or bound in hydroxyl-water groupings has little to no effect on T_g_ [8]. Water that disrupts interchain hydrogen bonds, however, depressed T_g_ significantly [36]. In general, water molecules plasticize polar polymers such as epoxy, and absorption of moisture can cause a significant reduction of T_g_. For epoxy resins, T_g_ is decreased on average by 20 K for each 1% of moisture absorbed [37]. Therefore, epoxy resins used for aerospace applications are highly cross-linked and may consist of rubber or nonpolar thermoplastic, which reduces the equilibrium moisture content and the drop in T_g_ [38].

## 5. Conclusions

The presented research was focused on the study of the chemical and thermal influence of the environment on the degradation of carbon composite with epoxy resin after different exposure times. Based on the obtained results, the following conclusions can be drawn:Based on the observation of the experimental CFRP composite surface macrostructure through light and electron microscopy, it can be concluded that exposure to 10% NaOH and 15% H_2_SO_4_ had the greatest impact on the CFRP composite. In contrast, exposure to 20% NaCl did not leave any macroscopically visible changes on the surface of the investigated CFRP composite.The results of tensile and three-point bending tests showed that the influence of chemical and thermal influences on the degradation of the CFRP composite led to a decrease in mechanical properties under most exposure conditions.Determination of the viscoelastic properties of the experimental material by DMA to obtain information on how the chemical environments of a given temperature affect the viscoelastic properties of the material after nine weeks of exposure confirmed a synergistic influence. It is conditioned by the chemical and temperature influence of the experimental environments which is caused by degradation changes and plasticizing effect due to the water absorption causing a reduction of modulus of elasticity.A more pronounced occurrence of two separation relaxation events appears only at a temperature of 60 °C. A higher peak with a higher T_g_ value represents the formation of a bond to the reinforcement, and a lower peak with a lower T_g_ value represents the degradation processes of the CFRP composite, which are caused by the cleavage of the polymer chain.

Based on the obtained research conclusions, further research can be interesting and necessary from the point of view of the behavior and degradation of CFRP composite in UV radiation and water solutions of calcium hydroxide, potassium hydroxide, hydrochloric acid, and phosphoric acids. That results can bring interesting information for comparison between the CFRP composites with carbon fibers and epoxy matrix, and GFRP composites with C glass fibers and polyester matrix which are of great importance for use in the chemical industry.

## Figures and Tables

**Figure 1 polymers-14-03245-f001:**
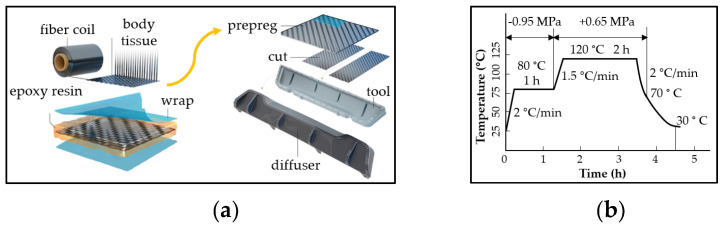
Production of the CFRP composite; (**a**) preparation flow chart; (**b**) curing cycle in the autoclave.

**Figure 2 polymers-14-03245-f002:**
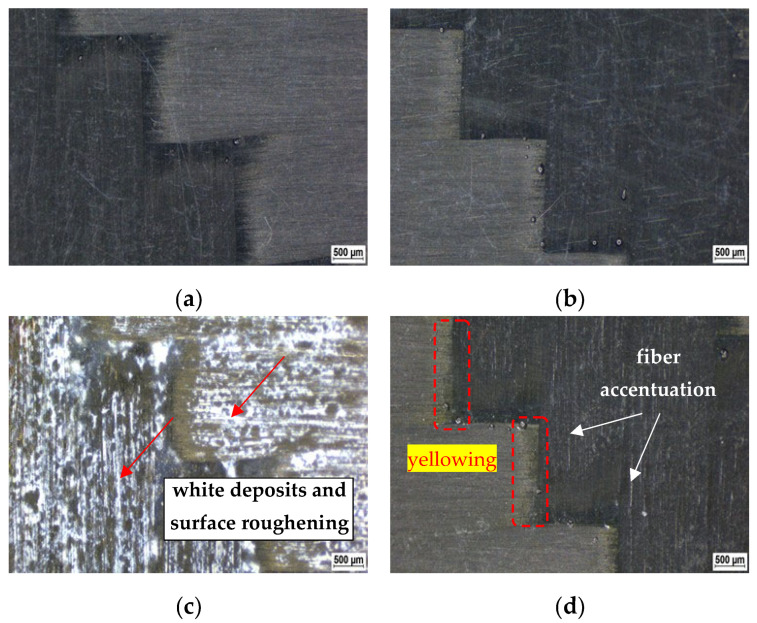
Material surface changes of the experimental CFRP composite; magnification 20× (after 9 weeks); (**a**) reference material before the test; (**b**) after 9 weeks of exposure to 20% NaCl at 60 °C; (**c**) after 9 weeks of exposure to 10% NaOH at 60 °C; (**d**) after 9 weeks of exposure to 15% H_2_SO_4_ at 60 °C.

**Figure 3 polymers-14-03245-f003:**
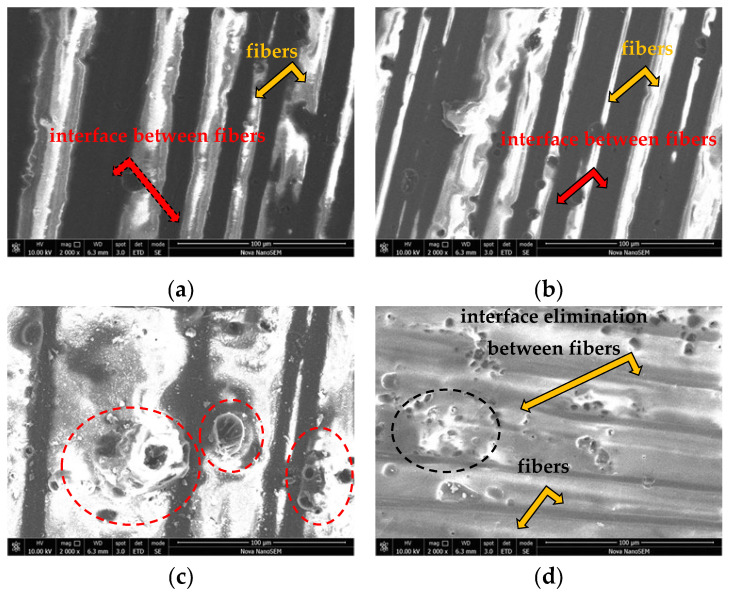
Material surface changes of the experimental CFRP composite; magnification 20× (after 9 weeks); (**a**) reference material before the test; (**b**) after 9 weeks of exposure to 20% NaCl at 60 °C; (**c**) after 9 weeks of exposure to 10% NaOH at 60 °C; (**d**) after 9 weeks of exposure to 15% H_2_SO_4_ at 60 °C.

**Figure 4 polymers-14-03245-f004:**
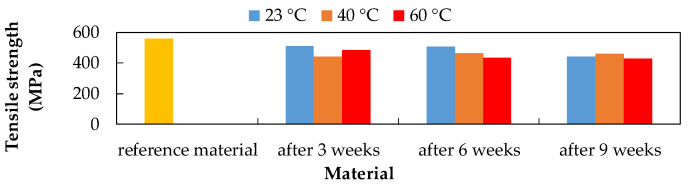
Influence of exposure to 20% NaCl at different temperatures on the CFRP composite.

**Figure 5 polymers-14-03245-f005:**
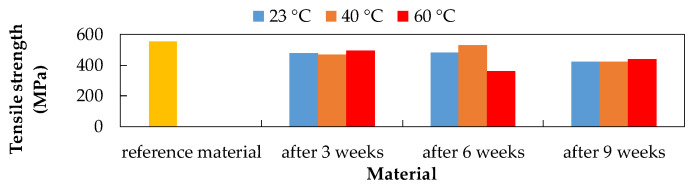
Influence of exposure to 10% NaOH at different temperatures on the CFRP composite.

**Figure 6 polymers-14-03245-f006:**
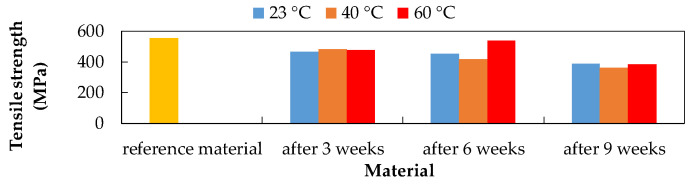
Influence of exposure to 15% H_2_SO_4_ at different temperatures on the CFRP composite.

**Figure 7 polymers-14-03245-f007:**
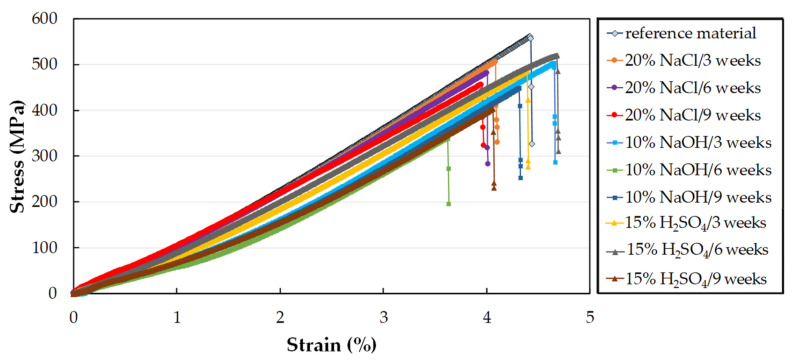
Influence of the time of exposure to 20% NaCl, 10% NaOH, and 15% H_2_SO_4_ at 60 °C on tensile stress-strain curves of the CFRP composite.

**Figure 8 polymers-14-03245-f008:**
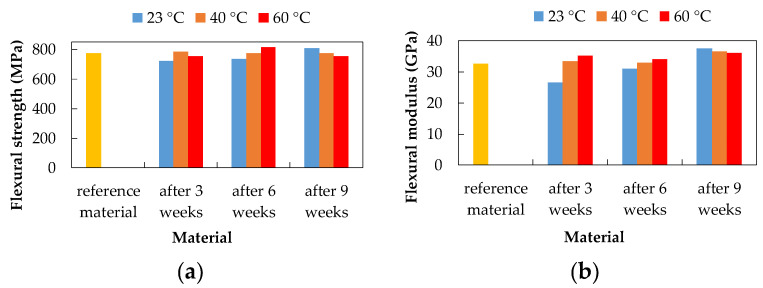
Influence of exposure to 20% NaCl at different temperatures on the CFRP composite; (**a**) flexural strength; (**b**) flexural modulus.

**Figure 9 polymers-14-03245-f009:**
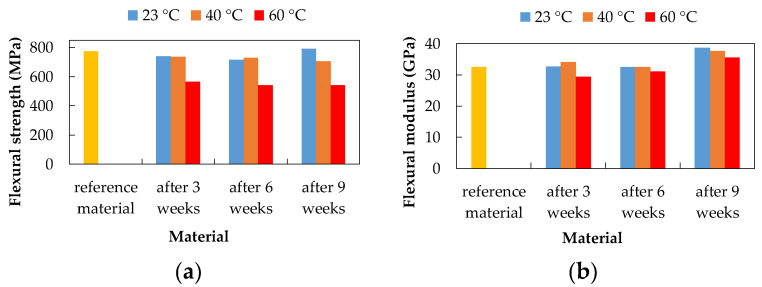
Influence of exposure to 10% NaOH at different temperatures on the CFRP composite; (**a**) flexural strength; (**b**) flexural modulus.

**Figure 10 polymers-14-03245-f010:**
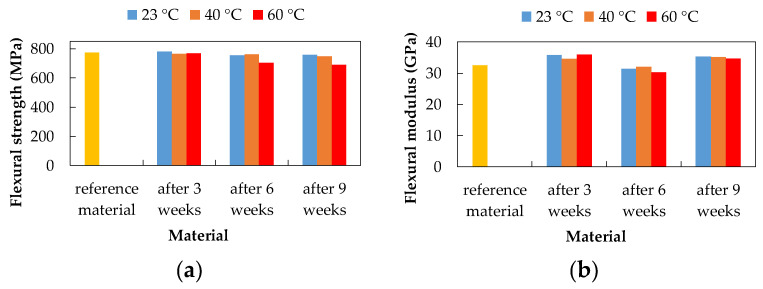
Influence of exposure to 15% H_2_SO_4_ at different temperatures on the CFRP composite; (**a**) flexural strength; (**b**) flexural modulus.

**Figure 11 polymers-14-03245-f011:**
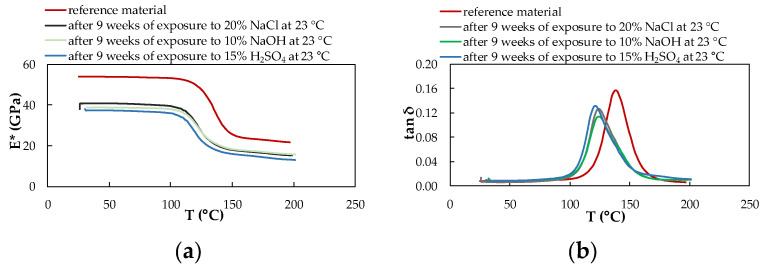
DMA of the CFRP composite evaluated depending on the influence of individual chemical environments at 23 °C; (**a**) complex modulus of elasticity; (**b**) tan delta.

**Figure 12 polymers-14-03245-f012:**
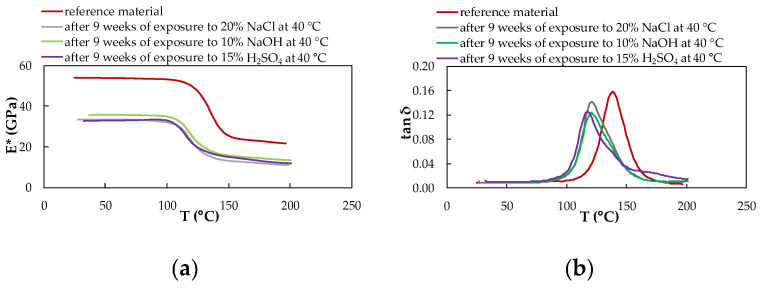
DMA of the CFRP composite evaluated depending on the influence of individual chemical environments at 40 °C; (**a**) complex modulus of elasticity; (**b**) tan delta.

**Figure 13 polymers-14-03245-f013:**
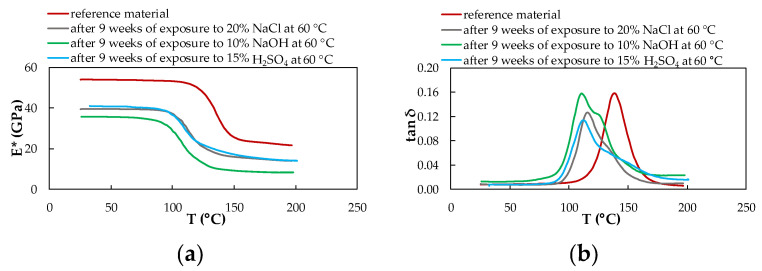
DMA of the CFRP composite evaluated depending on the influence of individual chemical environments at 60 °C; (**a**) complex modulus of elasticity; (**b**) tan delta.

**Table 1 polymers-14-03245-t001:** Examples of conducted research on the influence of chemical environments and humidity on composites.

Research	Material	Types of Conditioning	Variables	Methodology	Conclusions
[12]	GFRP	NaOH, HCl	alkaline and acid solution, time	three-point bending test	A decrease in flexural strength and flexural modulus over time, influence of NaOH base was more aggressive compared to HCl.
[14]	CFRP	water	temperature	immersion and mechanical properties	Degradation and decrease of mechanical properties.
[20]	GFRP	hydraulic fluid, or engine oil	types of environment	absorption, three-point bending test, impact test	A greater reduction in mechanical properties due to hydraulic fluid.
[21]	GFRP	salt fog and immersion into salt water	time of exposure and temperature	tensile test	GFRP laminates decreased their tensile strength after immersion in salt water.
[22]	CFRP and GFRP	water, salt water, alkaline environment, elevated temperature, exposure to diesel fuel	type of environment, type of FRP composite, time of exposure	tensile test, short beam shear test, and bending test	An adverse effect on the overall performance of CFRP and GFRP laminates, particularly bond strength, has been noted. Immersion in salt and alkaline environment significantly affect the GFRP samples.
[23]	GFRP	hygrothermal conditions	temperature	moisture absorption, tensile test, dynamic mechanical analysis, SEM (scanning electron microscopy), FTIR (Fourier-transform infrared spectroscopy)	Increased temperature accelerates the absorption of moisture and moisture diffusion coefficient. No influence was observed on the tensile strength and modulus of elasticity of the samples, with the exception of aging at 80 °C.
[24]	pultruded carbon/glass hybrid rod	underground oil well-exposure environment of elevated temperature hydraulic pressure and fatigue load	exposure conditions	the long-term evaluation of mechanical, and thermal properties and microstructure	The short beam shear strength and interface shear strength of rods decreased with the exposure time. Higher exposed temperature and hydraulic pressure aggravated the degradation by accelerating the diffusion of water molecules into HFRP rods.
[25]	basalt- and glass-fiber-reinforced polymer bars	seawater and sea sand concrete (SWSSC)	synergistic effect of sustained load and corrosion environment	scanning electron microscopy	SEM results reveal the degradation mechanism of BFRP bars in stress condition in SWSSC solution.

**Table 2 polymers-14-03245-t002:** Characteristics of experimental CFRP composite.

Layer	Orientation	Carbon Fiber Fabric	Epoxy Resin
1.	0/90°	GG280T(T300)	DT121HT-42 KE
2.	±45°		
3.	0/90°		

**Table 3 polymers-14-03245-t003:** Exposure conditions.

Weeks of Immersion	Chemical Solution (wt. %)	Temperature (°C)
3	20% NaCl	23
6	10% NaOH	40
9	15% H_2_SO_4_	60

**Table 4 polymers-14-03245-t004:** The glass transition temperature is determined by the dependence of the complex modulus of elasticity on temperature.

Chemical Solution	Temperature	T_g1_
-	-	122.95
20% NaCl	23	113.64
	40	107.95
	60	102.27
10% NaOH	23	110.80
	40	107.95
	60	93.75
15% H_2_SO_4_	23	107.95
	40	106.82
	60	96.59

**Table 5 polymers-14-03245-t005:** The glass transition temperature is determined by the dependence of the tan delta on temperature.

Chemical Solution	Temperature	T_g1_	T_g2_
-	-	138.24	
20% NaCl	23	120.59	
	40	123.53	
	60	116.28	126.47
10% NaOH	23	126.47	
	40	120.59	
	60	111.76	123.53
15% H_2_SO_4_	23	120.59	
	40	117.65	
	60	117.76	125.25

**Table 6 polymers-14-03245-t006:** Decrease in the tensile strength and flexural strength of the CFRP composite.

Mechanical Property	3 Weeks	6 Weeks	9 Weeks
Tensile strength	20% NaCl at 40 °C by 20.62%	10% NaOH at 60 °C by 35.13%	15% H_2_SO_4_ at 40 °C by 34.83%
Flexural strength	10% NaOH at 60 °C by 27.13%	10% NaOH at 60 °C by 29.97% 15% H_2_SO_4_ at 60 °C by 9.14%	10% NaOH at 60 °C by 29.96% 15% H_2_SO_4_ at 60 °C by 10.88%

## Data Availability

The data presented in this study are available on request from the corresponding author.

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
