# Peer review of "The Study of Chemical and Thermal Influences of the Environment on the Degradation of Mechanical Properties of Carbon Composite with Epoxy Resin"

_polymers, 2022, doi:10.3390/polym14163245_

Round 1

Reviewer 1 Report

This paper studies the Chemical and Thermal Influences of the Environment on the Degradation of Mechanical Properties of Carbon Composite with Epoxy Resin. The following major comments can further improve the quality of the paper. 

1# Abstract, please provide an analysis of degradation mechanisms of composite exposed in different corrosive environments. In addition, the three environments also have different effects on the degradation of composites. Please add relevant supplement.

2# The authors have summarized the properties evolution of FRP exposed to different environments. However, the summary of the degradation mechanism of long-term mechanical and thermal performance of FRP exposed to complex service environment is not sufficient, such as elevated temperature, hydraulic pressure and high salt concentration. Please review the latest research progress on FRP exposed to complex environments. Composite Structures, 2020; 246: 112418. Corrosion science, 2018, 138: 200-218.  International Journal of Fatigue, 2020, 134: 105480.

3# At the end of Part 1, please provide the main research work and the contribution and innovation of this paper.

4# If possible, please provide the preparation flow chart or product diagram of CFRP.

5# Please provide relevant selection basis on how to determine the weight percentage of different solution concentration in Table 3. What is the longest aging time? In addition, it is suggested to provide the selection basis of temperature, such as from references or relevant standards.

6# Please indicate the sample number for mechanical and thermal property tests. Another small suggestion can be provided to the authors. The two ends of CFRP plate are usually anchored with glued aluminum plates. However, the tensile strength obtained is lower than that of the material itself due to uneven or stress concentration as shown in Materials and Structures, 2018, 51:162. Therefore, it is suggested that the authors consider this in future research to avoid stress concentration and obtain the true tensile strength of CFRP.

7# On the material surface as shown in Figure 1, the authors can give some significant symbols and descriptions of words in the figure to indicate what actually happened on the surface before and after exposures? It is difficult for readers to capture some key information in current pictures.

8# At the end of 3.1, the authors claim the degradation of carbon fiber occurred in the exposure to 10% NaOH and 15% H2SO4. As known, carbon fiber is very stable when exposing alkaline/acid solution environments, so what is the degradation mechanism of carbon fiber?

9# Figures 3 to 5 show the degradation of tensile strength of CFRP in different environments. Why not provide the long-term evolution of tensile modulus and elongation at break? The latter two parameters are also crucial for practical engineering application and design.

10# At the end of part 3.3, it is suggested that the authors provide a quantitative comparison and analysis of mechanical properties in three different environments.

11# Why do two peaks appear in the green line after aging in Figure 11b? Please provide relevant explanations.

12# It can be seen from table 4 and 5 that with the increase of immersed temperature, the glass transition and temperature have a large decrease in three different solutions. However, according to the cognition of reviewer, the glass transition temperature should not change greatly in the salt solution environment, especially in the room temperature environment.

Author Response

Response to Reviewer 1 Comments

1# Abstract, please provide an analysis of degradation mechanisms of composite exposed in different corrosive environments. In addition, the three environments also have different effects on the degradation of composites. Please add relevant supplement.

The relevant supplement was added.

 2# The authors have summarized the properties evolution of FRP exposed to different environments. However, the summary of the degradation mechanism of long-term mechanical and thermal performance of FRP exposed to complex service environment is not sufficient, such as elevated temperature, hydraulic pressure and high salt concentration. Please review the latest research progress on FRP exposed to complex environments. Composite Structures, 2020; 246: 112418. Corrosion science, 2018, 138: 200-218.  International Journal of Fatigue, 2020, 134: 105480.

The summary of the degradation mechanism was improved.

3# At the end of Part 1, please provide the main research work and the contribution and innovation of this paper.

Folowing text was added: „In contrast with the composite literature, there are far few investigations that focus on the thermal and chemical influences on the degradation of mechanical properties of carbon fiber-reinforced polymer matrix composites [26,27]. To bring more information about short term behaviour of CFRP composites in the agressive industrial environmental conditions this study investigates the synergistic effect of environment and temperature on degradation of the CFRP composite produced from prepreg. The water solutions with higher concentrations (20% NaCl, 10% NaOH, and 15% H2SO4) and elevated temperature (23, 40 and 60 ˚C) were used to shorten the exposure time. The degradation changes of mechanical properties were studied in the time period of 3, 6 and 9 weeks using tensile test, three-point bending test, and dynamic mechanical analysis. The light microscopy and scanning electron microscopy (SEM) were used to investigate the degradation surface changes.“

4# If possible, please provide the preparation flow chart or product diagram of CFRP.

The flow chart was added, see Fig.1.

5# Please provide relevant selection basis on how to determine the weight percentage of different solution concentration in Table 3. What is the longest aging time? In addition, it is suggested to provide the selection basis of temperature, such as from references or relevant standards.

According to selection basis of temperature and concentrations of chemical solutions used in this study folowing text was added: „In technical practice, degradation changes of carbon composites with epoxy resin occur mostly in the temperature range of working temperatures of 20 – 90 °C. Some standards can be used in specific industrial conditions; for example, for testig of the CFRP composite in seawaterand sea sand concrete (SWSSC) conditions, but no international standards describe the testing conditions of the CFRP composite in alkaline and acidic water solutions. Many different testing conditions of immerson tests (duration, concentration and temperature) can be found in the literature. On the basis of demands of the CFRP composite producer and the short literature review [9,11-15,22,28,31,32] the exposure conditions listed in Tab. 3 were used.“

6# Please indicate the sample number for mechanical and thermal property tests. Another small suggestion can be provided to the authors. The two ends of CFRP plate are usually anchored with glued aluminum plates. However, the tensile strength obtained is lower than that of the material itself due to uneven or stress concentration as shown in Materials and Structures, 2018, 51:162. Therefore, it is suggested that the authors consider this in future research to avoid stress concentration and obtain the true tensile strength of CFRP.

The number of the samples used for mechanical tests was added in the text.

Thank you for valuable advice regarding the methodology of CFRP composites testing. We are going to consider this suggestion in our future tensile tests of CFRP plates to avoid stress concentration.

7# On the material surface as shown in Figure 1, the authors can give some significant symbols and descriptions of words in the figure to indicate what actually happened on the surface before and after exposures? It is difficult for readers to capture some key information in current pictures.

Symbols and descriptions were added in to the figures.

8# At the end of 3.1, the authors claim the degradation of carbon fiber occurred in the exposure to 10% NaOH and 15% H2SO4. As known, carbon fiber is very stable when exposing alkaline/acid solution environments, so what is the degradation mechanism of carbon fiber?

Sorry for the mistake. Sure, there is no degradation mechanism of carbon fiber.

9# Figures 3 to 5 show the degradation of tensile strength of CFRP in different environments. Why not provide the long-term evolution of tensile modulus and elongation at break? The latter two parameters are also crucial for practical engineering application and design.

Sure, tensile modulus and elongation at break can be also crucial for many practical engineering applications and mechanical design. But the CFRP composites are usually predominantly subjected to shear and bending mode of loading in the real applications. Therefore, flexural mechanical properties are in the center of interest very often when testing the CFRP laminates. Due to the request of the producer of real CFRP composite car components only the flexural modulus evolution was examined in this study.

10# At the end of part 3.3, it is suggested that the authors provide a quantitative comparison and analysis of mechanical properties in three different environments.

The comparison of degradation of mechanical properties in three various environments is shown in Table 6.

11# Why do two peaks appear in the green line after aging in Figure 11b? Please provide relevant explanations.

Two peaks appear after exposure in every selected chemical solution at temperature 60 °C, because the epoxy resin becomes more sensitive to degradation. Therefore, effect of temperature of 60 °C caused the biggest changes in material properties. In the green line after exposure to 10% NaOH at 60 °C is the greatest effect in comparison with 20% NaCl and 15% H2SO4 at 60 °C.

12# It can be seen from table 4 and 5 that with the increase of immersed temperature, the glass transition and temperature have a large decrease in three different solutions. However, according to the cognition of reviewer, the glass transition temperature should not change greatly in the salt solution environment, especially in the room temperature environment.

 You are right. It is expected that effect of exposure to 20% NaCl at 23 °C will have the smallest effect on material, however this wasn´t confirm by used dynamical analyze. It would be caused by high concentration of NaCl water solution. At lower concentrations the effect of exposure to NaCl at 23 °C on the glass transition temperature can be lower, probably.

Submission date

after review

2 August 2022

Reviewer 2 Report

See attached.

The manuscript, entitled “Study of Chemical and Thermal Influences of the Environment on the Degradation of Mechanical Properties of Carbon Composite with Epoxy Resin”, presented an experimental investigation on the effects of temperature, time, and chemicals on epoxy composite structures and properties. Overall, the authors showed adequate data, and a cohesive story. However, the authors need to improve the scientific soundness of this manuscript by revising some discussions. In the meantime, the authors also need to emphasize the motivation and the value of this work (in Introduction and Conclusions).

Major Points:

1.       Literature review:

There is nothing wrong with the content of the literature review part, but I have a few suggestions:

§  I would recommend moving the last paragraph (Line 121-129) ahead of the second last paragraph (Line 98-120), because the content in Line 121-129 is more similar to Line 85-94. In the current structure, the authors introduced a published paper (line 85-94), and then proposed the mechanism (Line 98-120), and then came back to another published paper result. It is clearer to have two paper reviews (Line 85-94 and Line 121-129) next to each other.

§  The authors reviewed many previously published works, and there is some apparent discrepancy. For example, in Line 76-77, ref.13, it is concluded that alkaline environment has more significant impact than acidic environment. In Line 128-129, ref. 23, it is concluded that acidic environment is more aggressive. It is totally understandable that there are many unmentioned experimental details that could possibly contribute to this difference. But the authors need to at least comment on this phenomenon to make it less confusing in Introduction part (I noticed that this was mentioned in Discussion section. Please point this out in Introduction)

2.       Experimental details:

§  Could the authors elaborate on the reasons for choosing these specific temperatures/chemical concentrations? (23  ͦC, 40  ͦC, 60  ͦC, 10%, 15%, 20%) The authors explained the temperature choice a bit in Discussion part. Please mention this in experimental part.

§  Please clarify the number of repeats for each tensile strength data point.

§  Could the authors provide an example of tensile loading curves? (One set of loading curve, possibly including unexposed, exposed at a certain chemical at 60 C for 3, 6 and 9 weeks) The tensile strength data could reflect some of the information, but the elongation, and tensile modulus information are missing from only showing tensile strength data. Having at least a set of data can be helpful for understanding what a typical tensile plot looks like.

3.       Discussion

§  Line 454-455, “chemical reactions between water molecules through hydrogen bonds.” Hydrogen bonds are not chemical reactions. The strength of hydrogen bonds is weak; thus, hydrogen bond is not chemical bond. Please modify this statement.

§  In the authors’ discussion section, chemical reaction (degradation) effect is emphasized. I think the authors need to raise the point that water acts at a plasticizer, reducing Tg by a lot even without any chemical reaction. Water absorption also reduces modulus due to plasticizing effect (physical reaction). More specifically, in the conclusion part, the authors drew the conclusion that viscoelastic change is due to degradation, which is not very accurate (Point 3, Conclusions).

4.       Conclusions

§  In the Conclusion section, the authors need to comment on the value of this research (how this work can guide the usage of materials in extreme conditions). The authors also mentioned that “Based on the obtained research conclusions, further research can be interesting and necessary from the point of view of the behavior and degradation of polymer composite materials in more aggressive environmental conditions.” This statement is vague. More aggressive environmental conditions can mimic more extreme applications or for scientific curiosity? The necessity and benefit of this future research need to be explained.

Minor Points: Grammars, Typos, etc.

§  The authors used a lot of complicated/long sentences. I would recommend the authors modify some of them to make the manuscript more readable.

§  Statement in Line 54, “As was also studied in many previous types of research”. The authors need to cite some previous research following this statement.

§  Section 1 is “Introduction”. The authors numbered another Section 1 “Literature review”. I think using subsection 1.1 “Literature review” will be better. Please correct.

§  Line 110-111: “Absorption of water by the resin can change the properties of the resin and the glass transition temperature of water molecules through hydrogen bonds.” Please correct this statement. Should be “glass transition temperature of polymers”

§  Line 269 “an 8.33 % decrese in the tensile strength of compared to the reference material” should be “an 8.33% decrease in the tensile strength compared the reference material”

§  Line 201 “Dynamic mechanical test (DMA)..”, Dynamic mechanical analysis, not dynamic mechanical test, is abbreviated as DMA. In Line 473-474, the authors mentioned “DMA analysis”. “DMA” is fine, which has included “analysis” in the word.

Author Response

Response to Reviewer 2 Comments

Major Points:

  1. Literature review:

There is nothing wrong with the content of the literature review part, but I have a few suggestions:

  • I would recommend moving the last paragraph (Line 121-129) ahead of the second last paragraph (Line 98-120), because the content in Line 121-129 is more similar to Line 85-94. In the current structure, the authors introduced a published paper (line 85-94), and then proposed the mechanism (Line 98-120), and then came back to another published paper result. It is clearer to have two paper reviews (Line 85-94 and Line 121-129) next to each other.

Accepted and corrected.

  • The authors reviewed many previously published works, and there is some apparent discrepancy. For example, in Line 76-77, ref.13, it is concluded that alkaline environment has more significant impact than acidic environment. In Line 128-129, ref. 23, it is concluded that acidic environment is more aggressive. It is totally understandable that there are many unmentioned experimental details that could possibly contribute to this difference. But the authors need to at least comment on this phenomenon to make it less confusing in Introduction part (I noticed that this was mentioned in Discussion section. Please point this out in Introduction)

Accepted and corrected.

Following text was added: „It cannot be clearly stated that due to the influence of an alkaline environment, there is a greater decrease in mechanical properties compared to an acidic environment, as was observed in the work of Mortas [23]. The effect of environment deeply depends on used concentration of acid and alkaline. But with comparison of the same concentration of acid and alkaline [12] the gratest decrease of material properties after alkaline was observed.“

  1. Experimental details:
  • Could the authors elaborate on the reasons for choosing these specific temperatures/chemical concentrations? (23  ͦC, 40  ͦC, 60  ͦC, 10%, 15%, 20%) The authors explained the temperature choice a bit in Discussion part. Please mention this in experimental part.

According to selection basis of temperature and concentrations of chemical solutions used in this study folowing text was added: „In technical practice, degradation changes of carbon composites with epoxy resin occur mostly in the temperature range of working temperatures of 20 – 90 °C. Some standards can be used in specific industrial conditions; for example, for testig of the CFRP composite in seawaterand sea sand concrete (SWSSC) conditions, but no international standards describe the testing conditions of the CFRP composite in alkaline and acidic water solutions. Many different testing conditions of immerson tests (duration, concentration and temperature) can be found in the literature. On the basis of demands of the CFRP composite producer and the short literature review [9-18] the exposure conditions listed in Tab. 3 were used.“

  • Please clarify the number of repeats for each tensile strength data point.

The number of the samples used for tensile tests and DMA was added in the text.

  • Could the authors provide an example of tensile loading curves? (One set of loading curve, possibly including unexposed, exposed at a certain chemical at 60 C for 3, 6 and 9 weeks) The tensile strength data could reflect some of the information, but the elongation, and tensile modulus information are missing from only showing tensile strength data. Having at least a set of data can be helpful for understanding what a typical tensile plot looks like.

 The example of typical stress-strain curves of the CFRP composite immersed in 20% NaCl, 10% NaOH and 15% H2SO4 at 60 °C for 3, 6 and 9 weeks was added (Figure 7).

  1. Discussion
  • Line 454-455, “chemical reactions between water molecules through hydrogen bonds.” Hydrogen bonds are not chemical reactions. The strength of hydrogen bonds is weak; thus, hydrogen bond is not chemical bond. Please modify this statement.
  • In the authors’ discussion section, chemical reaction (degradation) effect is emphasized. I think the authors need to raise the point that water acts at a plasticizer, reducing Tg by a lot even without any chemical reaction. Water absorption also reduces modulus due to plasticizing effect (physical reaction). More specifically, in the conclusion part, the authors drew the conclusion that viscoelastic change is due to degradation, which is not very accurate (Point 3, Conclusions).

Those statements in discusion section were modified as follows: „The reduction of tan delta curves is greater at elevated temperatures than at lower temper-atures of chemical environments. This might be due to the participation of the protons of the alkaline and acidic water solutions in chemical reactions between water molecules via hydrogen bonding and within the hydrophilic groups of the epoxy, which resulted in the generation of a less-plasticized region dispersed across the surface of the polymer [X]. According to previous studies, the irreversible degradation of aged materials with epoxy resin is due to moisture or water uptake, while plasticization occurs [23] and results in a decrease in the glass transition temperature [20, 24]. Delasi [124] proposes that the state of absorbed water has a significant effect on the associated variation in Tg. On the contrary, water contained in clusters or bound in hydroxyl-water groupings has little to no effect on Tg [8]. Water that disrupts interchain hydrogen bonds, however, depressed Tg signifi-cantly [40]. In general, water molecules plasticize polar polymers such as epoxy, and ab-sorption of moisture can cause a significant reduction of Tg. For epoxy resins, Tg is de-creased on average by 20 K for each 1% of moisture absorbed [Y]. Therefore, epoxy resins used for aerospace applications are highly cross-linked and may consist of rubber or nonpolar thermoplastic, which reduces the equilibrium moisture content and the drop in Tg [14].“

Point three in conclusion was corrected as follows: „Determination of the viscoelastic properties of the experimental material by DMA to obtain information on how the chemical environments of a given temperature affect the viscoelastic properties of the material after 9 weeks of exposure confirmed that a synergistic influence. It is conditioned by the chemical and temperature influence of the experimental environments which is caused by degradation changes and plasticizing effect due to the water absorption causing reduction of modulus of elasticity.“

  1. Conclusions
  • In the Conclusion section, the authors need to comment on the value of this research (how this work can guide the usage of materials in extreme conditions). The authors also mentioned that “Based on the obtained research conclusions, further research can be interesting and necessary from the point of view of the behavior and degradation of polymer composite materials in more aggressive environmental conditions.” This statement is vague. More aggressive environmental conditions can mimic more extreme applications or for scientific curiosity? The necessity and benefit of this future research need to be explained.

The conclusion was corrected.

Minor Points: Grammars, Typos, etc.

  • The authors used a lot of complicated/long sentences. I would recommend the authors modify some of them to make the manuscript more readable.
  • Statement in Line 54, “As was also studied in many previous types of research”. The authors need to cite some previous research following this statement.
  • Section 1 is “Introduction”. The authors numbered another Section 1 “Literature review”. I think using subsection 1.1 “Literature review” will be better. Please correct.
  • Line 110-111: “Absorption of water by the resin can change the properties of the resin and the glass transition temperature of water moleculesthrough hydrogen bonds.” Please correct this statement. Should be “glass transition temperature of polymers”
  • Line 269 “an 8.33 % decresein the tensile strength of compared to the reference material” should be “an 8.33% decrease in the tensile strength compared the reference material”
  • Line 201 “Dynamic mechanical test(DMA)..”, Dynamic mechanical analysis, not dynamic mechanical test, is abbreviated as DMA. In Line 473-474, the authors mentioned “DMA analysis”. “DMA” is fine, which has included “analysis” in the word.

Minor Points – it was corrected.

Submission date

after review

2 August 2022

Round 2

Reviewer 1 Report

It can be accepted. 

Reviewer 2 Report

The authors have done a great amount of work to improve the presentation quality and the scientific soundness. The authors also addressed my questions very well through revising this manuscript. I would recommend the acceptance of this manuscript.